# Relationship between Urban Floating Population Distribution and Livability Environment: Evidence from Guangzhou's Urban District, China

**Yang Wang** [1,*] **, Xiaoli Yue** [2,3] **, Hong'ou Zhang** [3] **, Yongxian Su** [3] **and Jing Qin** [4,5]

1. Faculty of Geography, Yunnan Normal University, Kunming 650500, China
2. School of Architecture and Urban Planning, Guangdong University of Technology, Guangzhou 510090, China; yxl199766@163.com
3. Key Lab of Guangdong for Utilization of Remote Sensing and Geographical Information System, Guangdong Open Laboratory of Geospatial Information Technology and Application, Guangzhou Institute of Geography, Guangdong Academy of Sciences, Guangzhou 510070, China; hozhang@gdas.ac.cn (H.Z.); suyongxian@gdas.ac.cn (Y.S.)
4. School of Tourism Sciences, Beijing International Studies University, Beijing 100024, China; qjing1986@163.com
5. Research Center of Beijing Tourism Development, Beijing 100024, China
* Correspondence: wyxkwy@163.com

**Abstract:** The livability environment is an important aspect of urban sustainable development. The floating population refers to people without local *hukou* (also called 'non-*hukou* migrants'). The floating population distribution is influenced by livability environment, but few studies have investigated this relationship. Especially, the influence of social environment on floating population distribution is rarely studied. Therefore, we study 1054 communities in Guangzhou's urban district to explore the relationship between livability environment and floating population distribution. The purpose of this article is to study how livability environment affects floating population distribution. We develop a conceptual framework of livability environment, which consists of physical environment, social environment and life convenience. A cross-sectional dataset of the impact of livability environment on the floating population distribution is developed covering the proportion of floating population in the community as the dependent variable, eight factors of livability environment as the explanatory variables, and two factors of architectural characteristics and one factor of location characteristics as the control variables. We use spatial regression models to explore the degree of influence and direction of physical environment, social environment and life convenience on the floating population distribution in livability environment. The results show that the spatial error model is more effective than ordinary least squares and spatial lag model models. The five factors of the livability environment have statistical significance regarding floating population distribution, including four social environment factors (proportion of middle- and high-class occupation population, proportion of highly educated people in the population, proportion of rental households, and unemployment rate) and regarding life convenience factors (work and shopping convenience). The conclusion has value for understanding how the social environment affects the residential choice of the floating population. This study will help city administrators reasonably guide the residential pattern of the floating population and formulate reasonable management policies, thereby improving the city's livability, attractiveness and sustainable development.

**Keywords:** livability environment; floating population; social environment; spatial regression model; Guangzhou's urban district

## 1. Introduction

Livability environment is an important aspect of urban sustainable development [1]. In "Transforming our World: The 2030 Agenda for Sustainable Development", 17 goals were

put forward. Goal 11 refers to the construction of "Sustainable Cities and Communities". The livability environment is closely related to this goal. Whether the urban floating population can tolerate the livability environment directly reflects whether the city has residential equity and environmental justice. Therefore, it is necessary to study the relationship between floating population distribution and the livability environment in cities.

The floating population represents people without local *hukou* (also called 'non-*hukou* migrants'). The floating population in China's megacities accounts for a large proportion of the population. The floating population is mainly composed of migrant workers (mainly engaged in manufacturing, construction and low-level service industries), migrant white-collar workers (mainly working in enterprises, government departments and public institutions) and new employment groups (such as newly graduated college students). The floating population is an important group to maintain cities' sustainable development. Compared to local residents, the characteristics of and mechanisms behind the floating population's residential choice are unique [2], and they are more easily influenced by concerned local institutional factors [3] and exclusive housing policies [4]. Therefore, these characteristics and mechanisms need to be studied.

Many factors affect residential location choice; among them, the livability environment is an important one. The physical environment, the social environment, and life convenience are all used to evaluate the livability environment in cities. Among them, many scholars have studied the influence of the physical environment and convenience factors on residential location choice. The factors in the physical environment mainly include: the built environment [5–8], positive landscape and environment [9–13] and negative landscape and environment [14–16]. Life convenience factors include public transportation convenience [17,18], school accessibility [19], employment accessibility [20] and workplace accessibility [21]. Compared with the above two types of factors, few studies have investigated the influence of social environment on residential location choice. In fact, the social environment is an important part of the livability environment in cities, and it is also decisive for residents when choosing thier housing location [22]. The Marxist, structuralist school and its social space unity theory hold that the social environment affects individual choice, which is an important dimension in the process of living decision-making, and individual choice also affects social space [23,24]. For areas with poor social environment, residents mostly adopt an evasive attitude [25]. To some extent, residential location choice can be regarded as the residents' choice of their social environment [26,27]. Residents tend to choose a neighborhood with similar social attributes to themselves (such as life background, occupation, economic level, educational level, social status, cultural level and ethnic characteristics) as their residence [28], and their neighborhood satisfaction is influenced by social environment characteristics [29]. Therefore, the social environment is an important starting point to study the residential location choice of floating population, which needs further attention.

This study focuses on the relationship between urban floating population distribution and livability environment and considers which factors of the livability environment have the most significant influence on floating population distribution. That is, among the physical environment, the social environment, and life convenience, which factors have the most obvious influence on the floating population? This has received little attention in previous studies. Moreover, there is a lack of studies on the influence of the social environment on urban floating population distribution.

Guangzhou is a highly developed city with a large proportion of floating population. According to the data of population sample survey in 2015, it is estimated that the floating population in Guangzhou accounts for 42.44% of its population. Guangzhou's physical environment [13], social environment [25,30–32] and life convenience [33] are complex and diverse and are very suitable for case studies.

Taking Guangzhou's urban district as an example, we build a conceptual framework of livability environment (including physical environment, social environment, and life convenience). On this basis, we construct a cross-sectional dataset that covers the proportion of

the floating population in 1054 communities in Guangzhou's urban district, eight factors of livability environment, two factors of architectural characteristics and one factor of location characteristics. We use a spatial regression model to explore the degree of influence of influence and direction of the impact of the physical environment, the social environment, and life convenience on the floating population distribution in livability environment. This will hopefully help city administrators to reasonably guide the residential pattern of the floating population and to formulate reasonable management policies to improve the city's livability, attractiveness and sustainable development capacity.

The rest of this paper proceeds as follows. Section 2 presents the conceptual framework of the livability environment, and our research design, indicators, data and research methods. Section 3 analyzes the degree of influence, and the direction, of the livability environment on the distribution of floating population in Guangzhou's urban district. Section 4 discusses the research findings and draws conclusions.

## 2. Concept and Methods

### 2.1. Conceptual Framework of Livability Environment on the Residential Location Choice of the Floating Population

The concept of livability environment is complex and diverse. In different research perspectives, scholars have presented different understandings of the concept. We present a conceptual framework of livability environment from the perspective of residential location choice of the floating population, which accounts for differences within urban communities. The framework includes three aspects: the physical environment, the social environment, and life convenience. Each aspect is evaluated by corresponding index (Figure 1).

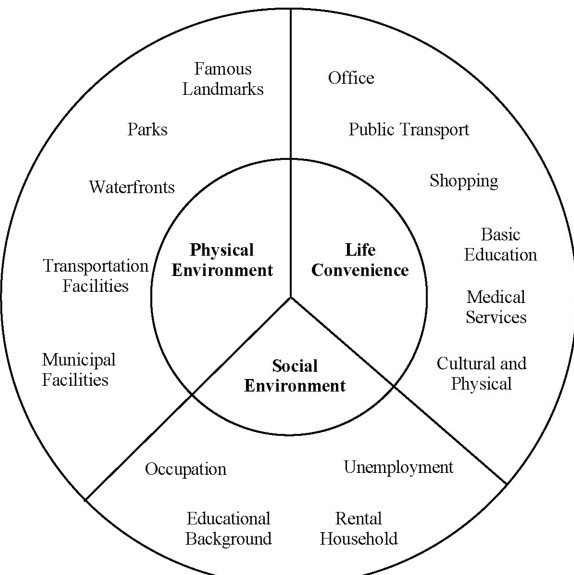

**Figure 1.** Illustration of livability environment on the residential location choice of the floating population.

#### 2.1.1. Physical Environment

Residents tend to prefer near-positive physical environments (NPPE) and avoid negative physical environments (ANPE) when choosing residential location [13]. In terms of NPPE, famous landmarks [34], parks [35] and waterfronts [36] have a positive effect on choice of residential location; as for ANPE, unwanted transportation facilities and municipal facilities have a negative impact on the living environment [14–16,37–40], and residents often want to stay away from the above facilities when they choose the location. Unwated transportation facilities include railways, highways, elevated roads, train stations, coach stations and airports, among others. Unwanted municipal facilities mainly include gas

stations, signal transmission towers, funeral homes, substations, high-voltage corridors, garbage disposal sites and sewage treatment plants, among others. These facilities may cause odor, dust, noise, radiation and other pollution to the surrounding environment or have a negative impact on mental health.

### 2.1.2. Social Environment

Social environment mainly includes occupation, educational background, rental household and unemployment. First, occupation can be denoted by the proportion of middle- and high-class occupations in the population (PMHCOP). In China, occupation classes can be divided into seven categories: (1) management, (2) professionals, (3) clerks and administrative staff, (4) workers in the retail and service sectors, (5) industrial workers, (6) workers in the agricultural sector and (7) unemployed; (1)–(4) can be defined as middle- and high-class occupations. Second, educational background can be denoted by the proportion of highly educated people in the population (PHEP); those with bachelor degree or above can be considered highly educated people. The percentage of highly educated people among those aged six and above is the indicator of a community's educational background. Third, in terms of rental household (denoted by the proportion of tenants in total households; PRH), Saunders' research shows that the ownership of residential property is the main determinant of social status [41]. Lockwood (2007) shows that the higher the proportion of rental housing in the community, the higher the violent crime rate [42]. Therefore, in theory, the higher the proportion of tenants in the community, the worse its security and social environment. Fourth, unemployment rate (UR) is an important index to measure the attractiveness and security of a community. Raphael and Winter–Ebmer (2001) show that the unemployment rate is positively correlated with the crime rate. Therefore, communities with high unemployment rate have relatively high risks of instability and insecurity, and people's income level is low, forming a poor social environment [43].

### 2.1.3. Life Convenience

Life convenience includes work and shopping convenience (WSC) and social public services convenience (SPSC). Among them, WSC includes office accessibility, public transport accessibility and commercial service accessibility, and the accessibility to office space, subway stations, and stores can be selected for evaluation. SPSC can be evaluated by the accessibility of basic education, medical services, and cultural and physical amenities. Previous studies have shown that the index factors of life convenience can significantly affect residents' residential location choices [44–46].

### 2.2. Study Area

Guangzhou is a Chinese megacity. The urban area of Guangzhou is the core and most important part of Guangzhou, having 379.71 km$^2$ and a permanent resident population of 5.96 million (Data of the Sixth Population Census in Guangzhou). Taking Guangzhou City as the research area, the scope is as follows: west to the boundary of Guangzhou City; south to the boundary of Haizhu District; east to Qianjin Sub-district Office, Huangcun Sub-district Office, Xintang Sub-district Office and Longdong Sub-district Office of Tianhe District; and north to Jingxi Sub-district Office, Tonghe Subdistrict Office, Huangshi Sub-district Office, Xinshi Sub-district Office, Tangjing Sub-district Office and Songzhou Sub-district Office of Baiyun District. The scope refers to the previous research results on the division of functional areas in Guangzhou [13]. From the perspective of administrative division, the urban area of Guangzhou includes Yuexiu District, Liwan District, Haizhu District, most of Tianhe District and the south of Baiyun District. Three communities in the study area have no data, so the research object is 1054 communities. The area can be further divided into old area, core area and marginal urban district (Figure 2). The Guangzhou's CBD is Zhujiang New Town, and the international finance center (IFC) is located at the center of the CBD.

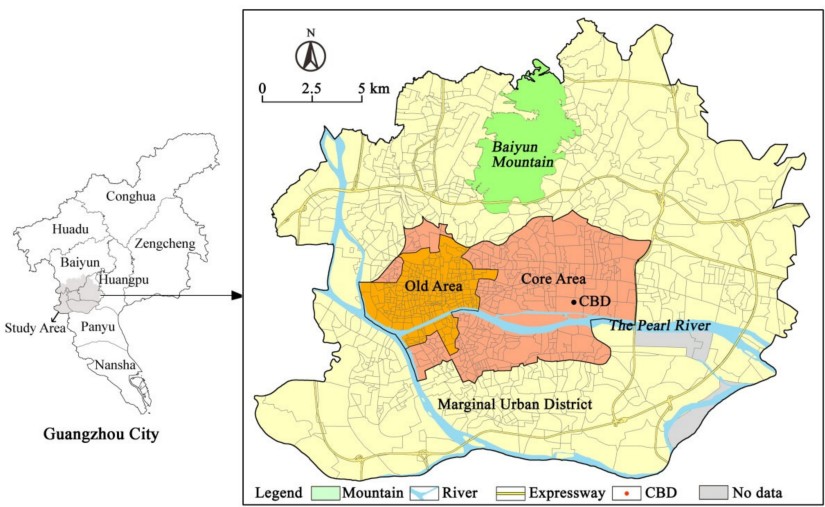

**Figure 2.** Study area.

*2.3. Research Design, and Indicator System and Model*

This study develops a research framework to analyze the influence direction and degree of livability environment on the floating population in the process of residential location selection. The analysis process is as follows: first, taking 1054 communities in Guangzhou as the research objects, the spatial distribution and spatial correlation characteristics of floating population were analyzed. Second, from the physical environment, social environment, life convenience and building and location characteristics, this paper constructs the residential choice model of floating population in Guangzhou urban area. Here, floating population is the dependent variable and physical environment, social environment and life convenience are explanatory variables, including eight indicators. There are three indicators for building and location characteristics, which are control variables. Third, we choose the appropriate model among ordinary least squares (OLS), spatial lag model (SLM) and spatial error model (SEM) to analyze the relationship between floating population and livability environment. This relationship includes significance, influence direction and influence intensity. Finally, the research results are analyzed (Figure 3).

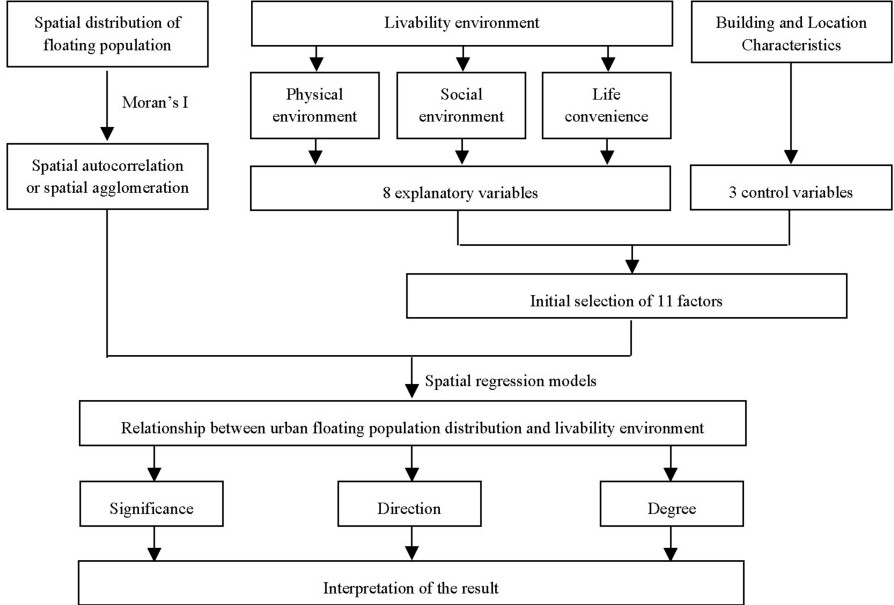

**Figure 3.** Research design.

The index system of decision-making factors of floating population living choice under livable orientation is constructed, as shown in Table 1.

**Table 1.** Definitions and evaluation methods for the variables.

| Variable (Symbol) | Evaluation Method or Index Composition | Expected Impact Direction |
|---|---|---|
| Dependent variable: Proportion of floating population (PFP) | PFP divided by total population. | |
| *Explanatory Variables—Physical Environment* | | |
| Near-positive physical environment (NPPE) | Comprehensive score for near famous landmarks, parks and waterfronts. | + |
| Avoid negative physical environment (ANPE) | Score for avoiding distasteful transportation facilities and municipal facilities. | + |
| *Explanatory Variables—Social Environment* | | |
| Proportion of middle- and high-class occupation population (PMHCOP) | middle- and high-class occupation population divided by the employed population; middle- and high-class occupations include (1) management (managers in government departments and other publicly funded agencies, nonprofit organizations, and corporations), (2) professionals, (3) clerks and administrative staff and (4) workers in the retail and service sectors. | + |
| Proportion of highly educated population (PHEP) | Undergraduate or above divided by population aged six years and above. | + |
| Proportion of rental households (PRH) | Rental household divided by total households. | - |
| Unemployment rate (UR) | Unemployed population divided by active people. | - |
| *Explanatory Variables—Life Convenience* | | |
| Work and shopping convenience (WSC) | Comprehensive score for the accessibility to office space, subway stations, and stores. | + |
| Social public services convenience (SPSC) | Comprehensive score for the conveniences of basic education, medical services, and cultural and physical amenities. | + |
| *Control Variables—Building and Location Characteristics* | | |
| Building age (BAGE) | Score of the average building age. | + |
| Building area per household (BAREA) | Score of the average building area per household. | + |
| Distance from the CBD (DCBD) | Distance from the IFC (km). | - |

According to Table 1, we construct a decision-making model of floating population living choice in Guangzhou urban area, which is used to analyze the relationship between urban floating population distribution and livability environment. The model is expressed as:

$$PFP = f \text{ (NPPE, ANPE, PMHCOP, PHEP, PRH, UR, WSC, SPSC, BAGE, BAREA, DCBD)} \tag{1}$$

Among them, PFP is the proportion of the population that is floating, which is a dependent variable of the model and represents the floating population status of the community. NPPE, ANPE, PMHCOP, PHEP, PRH, UR, WSC and SPSC are eight explanatory variables representing livability environment. BAGE, BAREA and DCBD are control variables.

BAGE and BAREA represent the building characteristics of the community. In theory, residents prefer to live in the houses of newer buildings and larger building areas [47–50]. DCBD is the most important indicator of location characteristics [13]. Location characteristics significantly affect residents' living choices [51].

*2.4. Data and Data Sources*

The data of PFP, PMHCOP, PHEP, PRH, UR, BAGE, and BAREA are derived from the data of the sixth census in Guangzhou. The relevant location data of NPPE, ANPE, WSC, SPSC, and DCBD are drawn according to POI point, line and area data of Baidu Map in 2012. Among them, the location data of distasteful municipal facilities in ANPE also refers to the corresponding current situation map of Guangzhou City Master Plan (2011–2020). Community boundaries are drawn according to the Atlas of Guangzhou Urban Management Community Network Responsibility Division.

*2.5. Methods*

2.5.1. Evaluation Method of Community Index Score

The calculation methods of PFP, PMHCOP, PHEP, PRH, UR and DCBD are shown in Table 1. To avoid large dimensional difference of data, we multiply PFP, PMHCOP, PHEP, PRH and UR by 100. See Wang et al. (2020) [13] for the scoring method of BAGE and BAREA. For the score calculation method of ANPE, see the evaluation method of "Avoid municipal facilities" by Wang et al. (2020) [13]. NPPE, WSC and SPSC are composite indicators; the score assignment methods of their sub indicators are shown by Wang et al. (2020) [13]. For the compound variables (NPPE, WSC and SPSC), the weights of the indicators are calculated through a factor analysis, and the weighted sum is employed to calculate their scores. Taking NPPE as an example, the NPPE score of a single community can be calculated as follows:

$$NPPE = \sum_{p=1}^{m} \left( w_p \times S_p \right) \tag{2}$$

In the formula, $S_p$ is the score value of the $p$-th sub index of NPPE, $m$ is the number of sub indexes and $w_p$ is the weight of the $p$-th sub index, which are calculated by factor analysis.

2.5.2. Analysis Method of Spatial Autocorrelation and Spatial Agglomeration of Foreign Population

When choosing residence, floating population often considers multiple nearby at the same time. Therefore, communities with similar proportion of floating population may have spatial agglomeration distribution and then produce spatial relevance. Global spatial autocorrelation index (GMI) is used to measure whether there is spatial autocorrelation characteristic of floating population distribution in Guangzhou's urban district.

2.5.3. Spatial Regression Model

To study the relationship between the distribution of floating population and livability environment in Guangzhou's urban district, we build a model of floating population's residential choice tendency. The model takes the proportion of floating population in the community as the dependent variable, eight indicators of livability environment in Table 1 as the explanatory variables (independent variables), and three indicators of building and location characteristics in Table 1 as the control variables. The OLS, SLM, and SEM are used to calculate, through comparison and selection, and the optimal model is selected to analyze, the relationship between the distribution of floating population and livability environment in Guangzhou's urban district.

OLS is a traditional linear regression model that can analyze the linear relationship between the proportion of floating population and 11 factors. The premise of the application of this model is that the factors of floating population's residential choice tendency are independent of each other and do not consider the spatial location relationship of the community. The expression of OLS model is as follows:

$$y_i = \beta X_i + \varepsilon_i, \left[ \varepsilon_i \sim N(0, \delta^2 I) \right] \tag{3}$$

In the above formula, $i = 1, 2, \ldots, 1054$, indicating the number of community samples in Guangzhou urban district; $y_i$ is the dependent variable and the proportion of floating population in the community; $X_i$ is the S-dimensional row vector ($s = 1, 2, \ldots, 11$) of the choice factors of the floating population, representing the value of the $s$-th influencing factor variable in the $i$-th community; $\beta$ is the S-dimensional column vector, which is the spatial regression coefficient corresponding to these 11 factors of residence choice of floating population; $\varepsilon$ is the error term of the model; and $\varepsilon_i \sim N(0, \delta^2 I)$ indicates that the error term obeys normal distribution and the variance is consistent, where $I$ is the identity matrix.

SLM, one form of spatial regression models, considers the influence of the proportion of floating population in a certain community on the proportion of floating population in other neighboring communities, that is, the spatial spillover effect. SLM can be expressed as [52,53]:

$$y_i = \rho \sum_{j=1}^{n} W_{ij} y_j + \beta X_i + \varepsilon_i, \left[ \varepsilon_i \sim N(0, \delta^2 I) \right] \tag{4}$$

where $\rho$ is the coefficient value of spatial autoregressive and $W_{ij}$ stands for spatial weight matrix.

SEM considers the possible spatial spillover benefits of independent error items in the model. SEM model is expressed as [52,54]:

$$y_i = \lambda \sum_{j=1}^{n} W_{ij} \varphi_i + \beta X_i + \varepsilon_i, \left[ \varepsilon_i \sim N(0, \delta^2 I) \right] \tag{5}$$

In the formula, $\varphi$ is the spatial autocorrelation error term in the model of residence choice of migrants and $\lambda$ is the spatial autocorrelation coefficient of the error term.

## 3. Results and Discussions

### 3.1. Spatial Difference Pattern of the Floating Population

Descriptive statistics of the proportion of floating population are shown in Figure 4. We create five data intervals: 0.00–20.00%, 20.01–40.00%, 40.01–60.00%, 60.01–80.00% and 80.01–97.44%. The number of communities in each interval is 459, 325, 121, 100 and 49, respectively. Figure 5 shows the spatial difference pattern of floating population distribution in Guangzhou city drawn by ArcGIS 10.0. Communities with over 40% floating population are mainly distributed in marginal urban district. Communities with less than 20% floating population are mainly distributed in old cities and core areas.

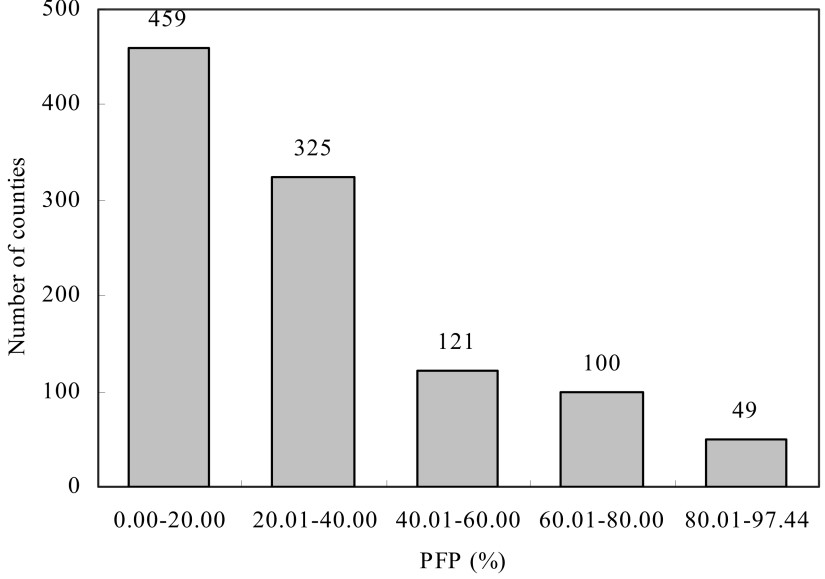

**Figure 4.** Descriptive statistics of floating population ratio.

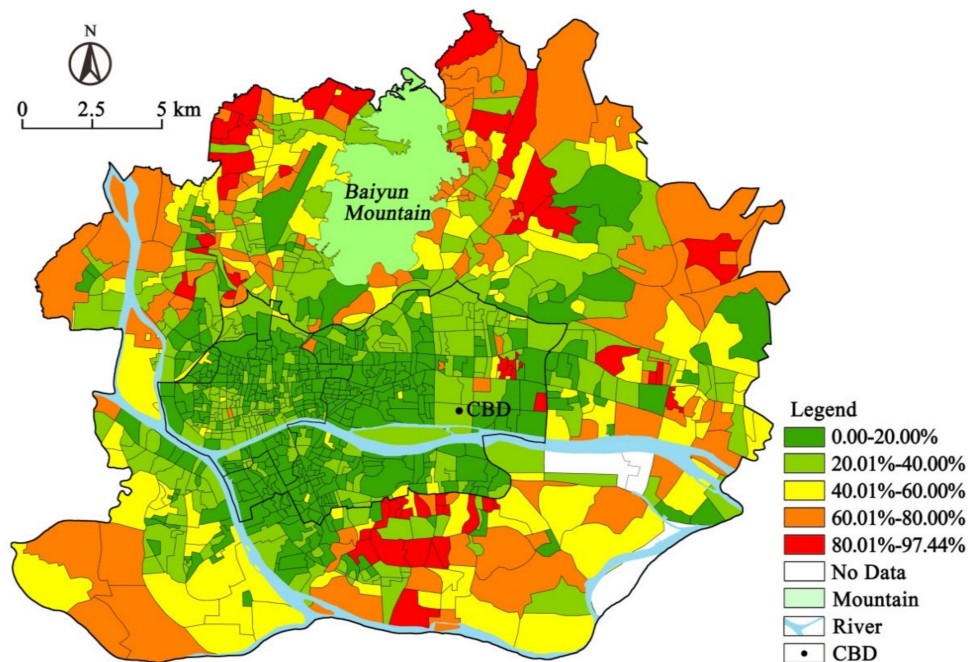

**Figure 5.** The spatial distribution of proportion of floating population in Guangzhou urban district.

Moran's I index was used to study the spatial autocorrelation characteristics of the proportion of floating population among 1054 communities in the region. The Moran's I index is 0.430952, the *p* value is 0.0000 and the Z statistical value is 29.8726, which shows that the proportion of migrants in Guangzhou's urban district presents significant spatial correlation characteristics. That is, the proportion of floating population in a certain community will be affected by the neighboring communities.

*3.2. Characteristics of Spatial Difference Pattern of the Livability Environment*

Eight livability environmental indicators (NPPE, ANPE, PMHCOP, PHEP, PRH, UR, WSC and SPSC) were classified into five categories by Nature Breaks (Jenks) (Figure 6). The higher the score, the better the livability environment. Communities with NPPE scores higher than 5.1534 are mainly distributed in the new central axis of Guangzhou, along the Pearl River and in the northern part of the old city. Communities with APPE scores below five are mainly distributed in peripheral urban areas. In terms of PMHCOP, the high value is mainly distributed in the core area and the low value is distributed in the southern part of the peripheral city. In terms of PHEP, the value in the east is generally higher than that in the west. The communities with PRH greater than 72.2629% are mainly distributed in the old city, and the PRH in the core area is generally lower than 29.3515%. The areas with UR higher than 11.9404 are mainly in the southwest of the old city and peripheral city. Communities with WSC scores greater than 6.9230 are concentrated in the old city and the area north of the Pearl River in the core area. However, the WSC scores of peripheral urban areas are generally lower than 2.3985. In terms of SPSC, the scores of the old city and the core area are generally higher than those of the peripheral city. In general, the spatial distribution of eight indicators of livability environment is heterogeneous. There are also differences in distribution patterns among the eight indicators.

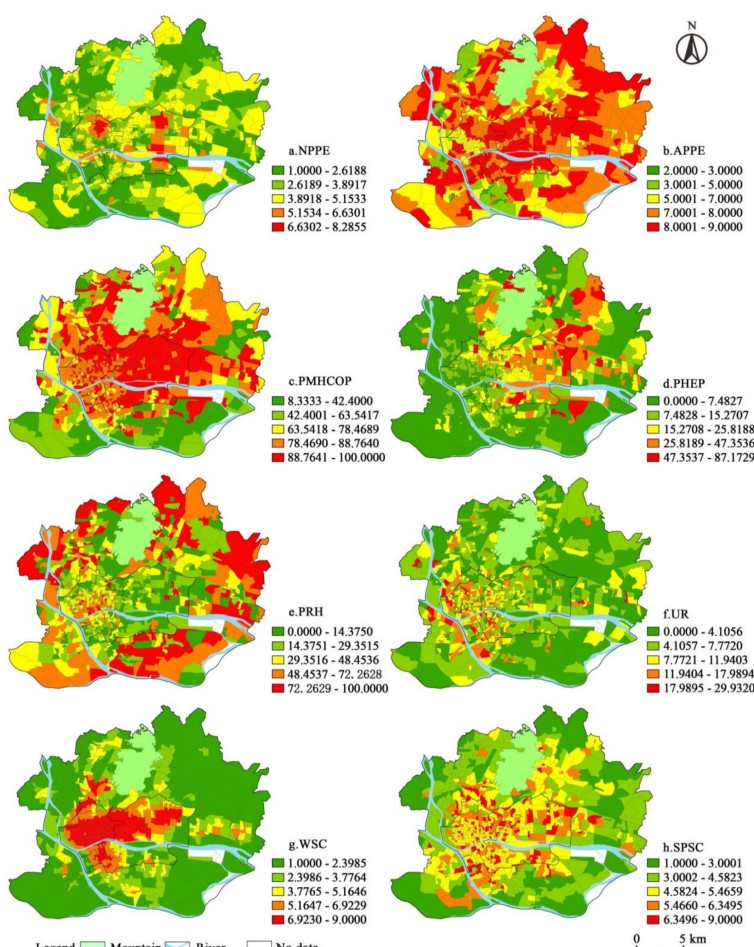

**Figure 6.** The spatial difference pattern of the livability environment in Guangzhou's urban district using the natural breaks (Jenks) method.

*3.3. Relationship between Urban Floating Population Distribution and Livability Environment in Guangzhou's Urban District*

First, the collinearity test is carried out on the 11 impact factors in Table 2 (Table 2). The test results show that the VIF values of all the 11 indicators are far lower than 10, and the factor with the largest VIF value (BAREA) is only 3.6461, which indicates that there is no obvious collinearity among these 11 factors, and all can be included in the regression model for factor analysis.

**Table 2.** The collinearity test of impact factors of floating population distribution in Guangzhou's urban district.

|  | Tolerance | VIF Value |
|---|---|---|
| NPPE | 0.8540 | 1.1709 |
| ANPE | 0.8875 | 1.1267 |
| PMHCOP | 0.5630 | 1.7761 |
| PHEP | 0.4700 | 2.1275 |
| PRH | 0.3545 | 2.8212 |
| UR | 0.7080 | 1.4123 |
| WSC | 0.5062 | 1.9755 |
| SPSC | 0.6332 | 1.5792 |
| BAGE | 0.4775 | 2.0944 |
| BAREA | 0.2743 | 3.6461 |
| DCBD | 0.6466 | 1.5465 |

The relationship between floating population distribution and livability environment was analyzed by three regression methods (OLS, SLM and SEM). The results (Table 3) show that among OLS, SLM and SEM, the $R^2$ of SEM model is the largest, reaching 0.769824, and AIC value is the lowest, reaching 8048.50. Thus, the fitting degree of the SEM is evidently better than the other two models, which again shows that the living choice of floating population in Guangzhou's urban district has significant spatial spillover effect. Therefore, the spatial error model is used to construct the decision-making model of floating population living choice in Guangzhou's urban district, and then the relationship between floating population distribution and livability environment is analyzed (Table 4).

**Table 3.** Comparison of the OLS, GWR and mixed GWR models.

| Model | $R^2$ | AIC | Log-Likelihood |
|---|---|---|---|
| OLS | 0.738748 | 8149.13 | −4062.57 |
| SLM | 0.758831 | 8072.03 | −4023.02 |
| SEM | 0.769824 | 8048.50 | −4012.25 |

**Table 4.** SEM model results.

| | Coefficient | Std. Error | t/z-Value | $p$ |
|---|---|---|---|---|
| NPPE | −0.2255 | 0.2759 | −0.8172 | 0.4138 |
| ANPE | −0.4869 | 0.3164 | −1.5391 | 0.1238 |
| PMHCOP | −0.1305 * | 0.0309 | −4.2173 | 0.0000 |
| PHEP | −0.1408 * | 0.0387 | −3.6406 | 0.0003 |
| PRH | 0.4379 * | 0.0222 | 19.7005 | 0.0000 |
| UR | −0.9470 * | 0.0786 | −12.0430 | 0.0000 |
| WSC | −1.0479 * | 0.2645 | −3.9621 | 0.0001 |
| SPSC | −0.2973 | 0.3368 | −0.8828 | 0.3774 |
| BAGE | 3.2891 * | 0.3771 | 8.7218 | 0.0000 |
| BAREA | −0.1287 | 0.7100 | −0.1812 | 0.8562 |
| DCBD | 0.3326 | 0.2503 | 1.3286 | 0.1840 |
| CONSTANT | 22.4242 * | 5.7040 | 3.9313 | 0.0001 |
| LAMBDA | 0.5028 * | 0.0458 | 10.9693 | 0.0000 |

$R^2$: 0.769824; AIC: 8048.50; Log likelihood: −4012.25

Note: * represent the 0.01 significance levels.

According to the significance of regression coefficient, we can judge the relationship between livability environment factors and the distribution of floating population. Table 4 shows that the four indicators of social environmental factors (PMHCOP, PHEP, PRH and UR) and WSC have a significant impact on the living choice of floating population. This shows that in the livability environment, social environment is the core factor affecting the distribution of floating population, while physical environment has no significant influence. In life community, work and shopping community has a significant impact on the distribution of floating population, while social public services community has no significant impact. The relationship between five significant livability environmental factors and the distribution of floating population is as follows.

In PMHCOP, every 1% increase in the community's promotion of middle-and high-class occupation population will reduce the proportion of floating population by 0.1305%. There is a negative correlation between them. It shows that the level of professional class is generally low in areas where the community floating population is concentrated and distributed. The floating population is at a disadvantage in the competition of professional class.

In PHEP, the community's proportion of highly educated population is negatively correlated with the proportion of floating population. Every 1% increase in the community's promotion of highly educated population will reduce the proportion of floating population in the community by 0.1408%. This shows that the education level of floating population is generally lower than that of registered population. It also shows that there are fewer highly educated talents in the floating population concentration areas.

In PRH, every 1% increase in community proportion of rental household will increase the proportion of floating population by 0.4379%. Renting a house is highly positively correlated with the floating population, which shows that the floating population mainly rents a house. Compared with the registered population, the floating population is at a disadvantage in obtaining housing property rights.

In terms of UR, the community's unemployment rate is negatively correlated with the proportion of floating population. For every 1% increase in community unemployment rate, the proportion of floating population will decrease by 0.9470%. This shows that the employment rate is higher in communities where floating population gather. Guangzhou is an employment-oriented immigrant city. The primary reason why the population immigrates to Guangzhou from other places and becomes a member of the floating population is that Guangzhou has more employment opportunities. If floating population is unemployed in Guangzhou, there is a high probability that it will move out of Guangzhou. Therefore, the employment rate of floating population is higher than that of registered population.

In WSC, the proportion of floating population in the community decreases by 1.0479% when the score of work and shopping convenience increases by one. This shows that the floating population often lives in communities with poor work and shopping convenience. Accessibility to office space, subway stations, and stores in communities with a large proportion of floating population are often poor. This may be due to their low-income level, and, thereby, poorer access to more expensive housing, making them choose to live in their area of work or shopping convenience.

## 4. Discussion and Conclusions

### 4.1. Discussion

This study presents the conceptual framework of livability environment guided by the living choice of floating population. Compared to the previous livability environment framework, the framework of this paper considers social environment factors, which are more suitable for explaining and analyzing the residential location choice of floating population in cities. At present, few studies have analyzed the living choices of floating population from the perspective of social space. In fact, social space is an important factor that cannot be ignored in the process of housing choice [22]. To some extent, choices of housing location can be regarded as their choice of social environment [26,27]. Residents tend to choose to live close to people with similar social attributes as their places of residence [28]. In the future, when studying the residential choice tendency of floating population in cities, we can adopt the theoretical framework proposed in this study and the perspective of dividing residential characteristics into "Physical Environment, Social Environment and Life Community" and select the corresponding indicators for research.

Based on the case study of Guangzhou's urban district, the relationship between the distribution of floating population and livability environment is described. The results show that the social environment is the most important factor for floating population's choice of residence. That is, the distribution of floating population is most closely related to social environment but not significantly related to physical environment. These findings have been rarely mentioned in previous studies. The results of this case study verify the rationality of the theoretical framework constructed in this paper. Researchers of Neo-Marxism and Structuralist School hold that understanding of social environment is an important factor in the process of residential decision-making [23,55]. Cassel and Mendelsohn (1985), who posit social space unity theory, believe that the choice of residential location has an interactive relationship with social space. For the above-mentioned basic viewpoints, this case study verifies the above-mentioned theory from the perspective of residence choice of floating population [24].

Notably, this article still has some limitations that need to be improved in the future. First, these livability environmental factors may have spatial heterogeneity effects on the distribution of the floating population. This study does not consider this effect. In the future, geographically weighted regression can be used to further analyze the spatial differences

of livability environmental factors. Second, different types of the floating population have different needs for living choices, so in the future, questionnaires or interviews can be used to analyze the characteristics of population differences. Third, using distance from the CBD to represent location characteristics is a way of simplifying assumptions. In the future, more refined methods and more indicators can be used to evaluate location characteristics.

### 4.2. Conclusions

In this study, a conceptual framework is established to analyze the relationship between floating population distribution and livability environment. From the perspective of residential location choice of the floating population, livability environment is divided into three aspects: physical environment, social environment and life convenience. A dataset of influencing factors of floating population's living choice in 1054 communities in Guangzhou city is established. Specifically, we focus on the possible degree of influence, and the direction, of physical environment, social environment and life convenience on the distribution of the floating population.

The research shows that the distribution of the floating population in Guangzhou has obvious spatial differences and spatial agglomerations. In general, the proportion of the floating population in the marginal urban district is the highest, while the proportion of the floating population in core area and old city is generally low. The livability environment in Guangzhou city presents spatial heterogeneity. Altogether, the livability environment in the core area is better, while that in the marginal urban district is relatively poor. There are differences in the spatial distribution pattern of livability environmental factors for eight aspects: NPPE, ANPE, PMHCOP, PHEP, PRH, UR, WSC and SPSC.

In this study, SEM is used to test the relationship between the distribution of floating population and livability environment and the significance of this relationship. The results show that five factors of livability environment have statistical significance on the distribution of floating population. The five factors include four social environment factors (PMHCOP, PHEP, PRH and UR) and one life community factor (WSC). Based on the relationship between PMHCOP, PHEP, PRH and WSC and the proportion of floating population, it shows that floating population are concentrated in communities with poor livability environment, which is consistent with theoretical expectations. Only the relationship between UR and the distribution of floating population is inconsistent with the theoretical expectation. This can be explained by the employment-oriented inflow of the floating population.

**Author Contributions:** Conceptualization, Y.W.; methodology, X.Y. and J.Q.; formal analysis, Y.W., X.Y. and Y.S.; writing—original draft preparation, Y.W. and H.Z.; writing—review and editing, Y.W., X.Y. and H.Z.; Visualization, Y.W. All authors have read and agreed to the published version of the manuscript.

**Funding:** This research was funded by the National Natural Science Foundation of China (No. 41871150, 41801168); GDAS Special Project of Science and Technology Development (No. 2020GDASYL-20200104001; 2021GDASYL-20210103004); Special Project of Institute of Strategy Research for Guangdong, Hong Kong, and Macao Greater Bay Area Construction (No. 2021GDASYL-20210401001); and National Key Research and Development Program (No. 2019YFB2103101).

**Institutional Review Board Statement:** Not applicable.

**Informed Consent Statement:** Not applicable.

**Conflicts of Interest:** The authors declare no conflict of interest.

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
