# Peer review of "Relationship between Urban Floating Population Distribution and Livability Environment: Evidence from Guangzhou’s Urban District, China"

_sustainability, doi:10.3390/su132313477_

Round 1
Reviewer 1 Report
Τhe current form of the paper is of high standards
Author Response
Point 1: The current form of the paper is of high standards.
Response 1:We are thankful for your positive comment.

Reviewer 2 Report
Thank you for giving me the opportunity to review the article „Relationship between urban floating population distribution and livability environment: Evidence from Guangzhou's urban district, China”. Below my remarks:
- the Authors have chosen an interesting and current topic on relationship between urban floating population distribution and livability environment;
- I think that the abstract and the introduction lack a clear purpose for the article;
- the literature review prepared by the Authors is correct;
- the Authors have designed the research process correct. Authors construct a cross-sectional dataset that covers the propor-tion of the floating population in 1054 communities in Guangzhou’s urban district, eight factors of livability environment, two factors of architectural characteristics and one factor of location characteristics;
- the Authors described the results of the research and used appropriate methods to analyze them;
- In the discussion, the Authors referred to results obtained by other researchers;
- the article is technically carefully prepared.
Congratulations to the authors on an interesting article.
Author Response
Point 1: I think that the abstract and the introduction lack a clear purpose for the article.
Response 1:We are thankful for your suggestions. We have added the purpose for the article in the abstract as follows:
The purpose of this article is to study how the livability environment affects the floating population distribution.
Point 2: the literature review prepared by the Authors is correct;
the Authors have designed the research process correct. Authors construct a cross-sectional dataset that covers the proportion of the floating population in 1054 communities in Guangzhou's urban district, eight factors of livability environment, two factors of architectural characteristics and one factor of location characteristics;
the Authors described the results of the research and used appropriate methods to analyze them;
In the discussion, the Authors referred to results obtained by other researchers;
the article is technically carefully prepared;
Congratulations to the authors on an interesting article.
Response 2:We are thankful for your positive comment.

Reviewer 3 Report
Dear Authors,
I find the article prepared by you very good and worth publishing in Sustainability. Both the editing and the content are of a high standard. Among the merits I notice above all the detailed explanation of terms important for the research (e.g. floating population, livability environment etc.) as well as their empirical operationalization. In principle, as a reviewer I could agree to publish the article in its current form. However, I have three minor comments, the inclusion of which will improve your article.
First, the summary could be a bit shorter but still include one sentence explaining what the title "floating population" is. In the second paragraph of Intoduction I find such a sentence.
Second, on page 9 in the last paragraph you wrote that "In theory, residents tend to live in areas close to CBD". I have trouble interpreting this sentence, and perhaps future readers of your article will too. Are you referring to your research assumption that residents prefer to live close to CBDs? Or do you mean the specific scientific theory or concept that says residents prefer to live close to the CBD? If it is the second option, then please provide the name of the theory (possibly the author) and the source. I note that the term CBD, is based on the work of the Alonso-Muth-Mils, who are the founders of Natural Evolution Theory. The theory explains that when employment is concentrated in a CBD, residential location is placed outside areas of employment concentration. Jobs are clustered in the CBD, and adjacent areas are primarily allocated for business use. As the city grows, commercial buildings are located in more neighborhoods and residential areas are pushed outward. The growth of new residential development on the periphery, results in the migration of wealthier segments of society to these areas. Old urban neighborhoods are left to lower-income social groups. The result is a stratification of society in the urban area and residents do not at all pursue locations close to the CBD. Lower-income social groups remain in the city, while wealthier households settle in the suburbs. Furthermore, the question arises, is Guangzhou a city that is developing based on the CBD concept? I did not find such a statement in the article - perhaps the authors would refer to the city's spatial plan? Thus, in my opinion, the problematic sentence is too much of a "mental shotrcut" by the authors and should be clarified.
Third, it would be good to add limitations of the study in the Conclusion. I mean one paragraph explaining the simplifying assumptions (e.g., regarding CBD), data deficiencies, etc. Your article is good enough to be used by other authors in the future. It would be useful for these authors to be aware of these limitations and to be able to more accurately place your conclusions in their considerations.
Author Response
Point 1: The summary could be a bit shorter but still include one sentence explaining what the title "floating population" is. In the second paragraph of the Introduction, I find such a sentence.
Response 1: We are thankful for your suggestions. We have added one sentence explaining what the title "floating population" is in the abstract. Please find the detailed revisions in the revised manuscript.
The floating population is the person without local hukou (also called ‘non-hukou migrants’).
Meanwhile, we deleted two sentences in the abstract to reduce the number of words in the abstract.
Point 2: On page 9 in the last paragraph you wrote that "In theory, residents tend to live in areas close to CBD". I have trouble interpreting this sentence, and perhaps future readers of your article will too. Are you referring to your research assumption that residents prefer to live close to CBDs? Or do you mean the specific scientific theory or concept that says residents prefer to live close to the CBD? If it is the second option, then please provide the name of the theory (possibly the author) and the source. I note that the term CBD, is based on the work of the Alonso-Muth-Mils, who are the founders of Natural Evolution Theory. The theory explains that when employment is concentrated in a CBD, residential location is placed outside areas of employment concentration. Jobs are clustered in the CBD, and adjacent areas are primarily allocated for business use. As the city grows, commercial buildings are located in more neighborhoods and residential areas are pushed outward. The growth of new residential development on the periphery, results in the migration of wealthier segments of society to these areas. Old urban neighborhoods are left to lower-income social groups. The result is a stratification of society in the urban area and residents do not at all pursue locations close to the CBD. Lower-income social groups remain in the city, while wealthier households settle in the suburbs. Furthermore, the question arises, is Guangzhou a city that is developing based on the CBD concept? I did not find such a statement in the article-perhaps the authors would refer to the city's spatial plan? Thus, in my opinion, the problematic sentence is too much of a "mental shotrcut" by the authors and should be clarified.
Response 2: Special thanks for your suggestions. The expression that “In theory, residents tend to live in areas close to CBD” is really wrong. We have deleted this sentence.
Point 3: It would be good to add limitations of the study in the Conclusion. I mean one paragraph explaining the simplifying assumptions (e.g., regarding CBD), data deficiencies, etc. Your article is good enough to be used by other authors in the future. It would be useful for these authors to be aware of these limitations and to be able to more accurately place your conclusions in their considerations.
Response 3:Special thanks for your suggestions. We have added limitations of the study in the discussion and conclusion section. Please find the detailed revisions in the revised manuscript.
Notably, this article still has some limitations that need to be improved in the future. First, these livability environmental factors may have spatial heterogeneity effects on the distribution of the floating population. This study does not consider this effect. In the future, geographically weighted regression can be used to further analyze the spatial differences of livability environmental factors. Second, different types of the floating population have different needs for living choices, so in the future, questionnaires or interviews can be used to analyze the characteristics of population differences. Third, using distance from the CBD to represent location characteristics is a way of simplifying assumptions. In the future, more refined methods and more indicators can be used to evaluate location characteristics.

Reviewer 4 Report
I find this to be an in-depth study worth publishing - it is overall an appropriate and consistent research. However I would recommend to broaden the conclusions of this research work, and to include proposed further studies to outweight the limitations of the current work.
Thank you in advance.
Author Response
Point 1: I find this to be an in-depth study worth publishing-it is overall an appropriate and consistent research. However I would recommend to broaden the conclusions of this research work, and to include proposed further studies to outweigh the limitations of the current work.
Thank you in advance
Response 1:Special thanks for your suggestions. We have added limitations and possible future research directions in discussion and conclusion section. Please find the detailed revisions in the revised manuscript.
Notably, this article still has some limitations that need to be improved in the future. First, these livability environmental factors may have spatial heterogeneity effects on the distribution of the floating population. This study does not consider this effect. In the future, geographically weighted regression can be used to further analyze the spatial differences of livability environmental factors. Second, different types of the floating population have different needs for living choices, so in the future, questionnaires or interviews can be used to analyze the characteristics of population differences. Third, using distance from the CBD to represent location characteristics is a way of simplifying assumptions. In the future, more refined methods and more indicators can be used to evaluate location characteristics.
